# EMERGENT COMMUNICATION FINE-TUNING (EC-FT) FOR PRE-TRAINED LANGUAGE MODELS

**Shane Steinert-Threlkeld**
Department of Linguistics
University of Washington
shanest@uw.edu

**Xuhui Zhou**
Language Technologies Institute
Carnegie Mellon University
xuhuiz@andrew.cmu.edu

**Leo Z. Liu**
Paul G. Allen School of Computer Science and Engineering
University of Washington
zeyuliu2@cs.washington.edu

**C.M. Downey**
Department of Linguistics
University of Washington
cmdowney@uw.edu

## ABSTRACT

It has been argued that the currently dominant paradigm in NLP of pretraining on text-only corpora will not yield robust natural language understanding systems. One strain of this argumentation highlights the need for grounded, goal-oriented, and interactive language learning. In this position paper, we articulate how **E**mergent **C**ommunication (EC) can be used in conjunction with large pre-trained language models as a '**F**ine-**T**uning' (FT) step (hence, EC-FT) in order to provide them with supervision from such learning scenarios. We discuss methodological issues and difficulties with making this work, and then illustrate the overall idea with a case study in unsupervised machine translation, before concluding with a discussion on the relation to multimodal pretraining.

## 1 EC-FT: MOTIVATION

Recent breakthroughs in myriad NLP tasks and datasets have been driven by large-scale pretraining from raw text corpora using semi-supervised learning signals (Howard & Ruder, 2018; Peters et al., 2018; Devlin et al., 2019; Liu et al., 2019; Conneau et al., 2020; Liu et al., 2020; Brown et al., 2020, i.a.).Despite these breakthroughs, several recent results suggest limitations on their generalization performance (McCoy et al., 2019; Niven & Kao, 2019; Ettinger, 2020; Rogers et al., 2020, i.a.). More fundamentally, a number of recent papers have argued that pretraining on text alone will not deliver fully general and robust NLP systems. For example, through several detailed thought experiments, Bender & Koller (2020) argue that models trained on text alone, in principle, will not be able to recover either the conventional/standing meaning of expressions or the communicative intent of an expression in context. Many of their arguments highlight the importance of the interaction between linguistic expressions and extra-linguistic context communicative intents (e.g. acting in the world, executing programs, etc.).[1] Similarly, Bisk et al. (2020) articulate progressively broader *world scopes* in which language use is embedded, and argue that present pretraining methods work at a relatively limited scope. They too emphasize the importance of embodied interaction with the environment and with the social world for future NLP systems.[2]

At the same time, the growing field of emergent communication (Wagner et al., 2003; Skyrms, 2010; Lazaridou & Baroni, 2020) studies artificial agents communicating with each other in order to accomplish particular goals in an environment. This can be viewed as a subfield of reinforcement learning, wherein language (i.e. the communcation protocol) is shaped by rewards determined by

---

[1] See Merrill et al. (2021) for a formalization of Bender & Koller (2020)'s argument about learning a programming language from form alone.

[2] As noted by Bender & Koller (2020), many of these arguments can be seen as detailed elaborations of the need for NLU systems so solve the *symbol grounding* problem Harnad (1990); Taddeo & Floridi (2005). We thank an anonymous reviewer for highlighting this connection.

interacting with an external environment and with other agents. Typical work in this area starts, however, from a *tabula rasa* and studies under what conditions—e.g. environments, tasks/goals, social settings—the resulting communication protocols among agents resembles human language, along axes like word length economy (Chaabouni et al., 2019a), word-order biases (Chaabouni et al., 2019b), and compositionality (Chaabouni et al., 2020; Steinert-Threlkeld, 2020; Geffen Lan et al., 2020), among others (Mu & Goodman, 2021).

In this short position paper, we articulate a vision for using the goal-directed, interactive setting of emergent communication as a method for fine-tuning pretrained language models. This EC-FT approach has the potential to combine the strengths of large-scale pretraining with the benefits of grounded, goal-directed communication to build better NLP systems. We first (§ 2) articulate some implementation choices for executing an EC-FT project before illustrating them with a case study on unsupervised machine translation (§ 3) and finally discussing related (§ 4) and future (§ 5) work.

## 2 IMPLEMENTATION CHOICE POINTS

Executing EC-FT is non-trivial: many issues arise when integrating text-only models into a larger, interactive scenario. We here outline some of the choice points, focusing on pieces that are unique to embedding a pretrained model in an EC setting as opposed to EC in general.

**Type of pretrained model:** while this choice will partially be determined by downstream task desires, we note that a primary choice for transformers comes from whether to use an encoder-only (e.g. BERT/RoBERTa), decoder-only (e.g. the GPT series) or an encoder-decoder model (e.g. (m)BART (Lewis et al., 2020; Liu et al., 2020) and MASS (Song et al., 2019)). Given that EC effectively 'inverts' sequence-to-sequence tasks, with a sender (a decoder) generating a message that's consumed by a receiver (an encoder), it is natural to use pretrained models with both components.[3]

**Environment-to-sender adapter:** Depending on the choice of EC game, some extra-linguistic environment will need to be presented to a sender pretrained as a text-only decoder. For example , in a traditional image reference game (Lazaridou et al., 2017; Havrylov & Titov; Lee et al., 2018, i.a.), this will involve presenting one or several images to the sender. Two special considerations arise in the EC-FT setting. First, images are often represented by extraction from a pretrained vision model, and then applying a small learned transformation to the sender's embedding space. In the EC-FT setting, we are primarily interested in training the language-modeling components and exporting them to other language-only tasks. This provides some *prima facie* reason to prefer fixed, non-learnable transformations of image representations into the sender's embedding space. Candidates include various strided pooling operations or other dimensionality reduction techniques like PCA.

Second, an issue arises when the sender is initialized from a pretrained transformer decoder (as will be common in the current era), which heavily relies on cross-attention to a set of representations (typically from an encoder). When using recurrent networks, a fixed image representation can be "presented" to the sender by supplying it as the initial hidden or cell state. Here, however, the environment needs to be represented not just as one representation, but as a sequence for the purposes of cross-attention.[4] In the image reference game scenario, this can be accomplished with a small LSTM that takes an image representation and generates a sequence, or via more sophisticated region-of-interest extraction methods as used in vision transformers (Li et al., 2019b; Tan & Bansal, 2019; Du et al., 2022). Intuitively, inputs to the cross-attention of the sender/decoder should occupy a similar space as the encodings that the model was pre-trained with, so as to mitigate catastrophic forgetting. For this reason, a learned transformation may be more appropriate than a fixed one.

**Receiver-to-environment adapter:** On the other end of the pipeline, a transformer encoder receiver takes the sender's natural language generation as input and produces a set of representations rather than a single one. Thus, a pooling or aggregation step is necessary. This could involve using a

---

[3]This is, of course, not necessary. For instance, XLM(-R) (Conneau et al., 2020) is an encoder-only model that was fine-tuned for sequence-to-sequence tasks by initializing a decoder with the same weights.

[4]There are at least two reasons that an environment (e.g. image) embedding cannot simply be the first token to the sender. (1) Many pretrained models use the first token as a special control code (e.g. a language ID in multingual models (Liu et al., 2020)). (2) It is not obvious that a transformer decoder's cross-attention can be 'turned off' without effecting overall performance, since the model has always been trained with it.

special position token (e.g. `[CLS]`) as the sentence representation, mean/max pooling, or learning a light aggregator over the final representations (e.g. a small recurrent network).

**Drift / forgetting:** it has been observed in other contexts that EC can cause significant language drift (Lee et al., 2018; 2019; Lu et al., 2020; Lazaridou et al., 2020). This is likely to be particularly acute in the EC-FT setting, where the pretraining tasks are very different from typical EC tasks. Methods such as KL regularization (Havrylov & Titov; Baziotis et al., 2020) are likely to be helpful here, as are other techniques from the catastrophic forgetting literature (Parisi et al., 2019).

## 3 CASE STUDY: UNSUPERVISED TRANSLATION

We here describe an ongoing project which aims to build better systems for *unsupervised* neural machine translation (UNMT) by taking large pretrained multilingual models and embedding them in an EC game, i.e. having them partake in goal-directed communication. At a high level, communication should promote translation in the following way. The task of translation can be viewed as 'aligning' a model's representations for sentences in several languages. In the supervised case, the parallel text data instructs the model how to do this alignment. In the unsupervised case, there's no such direct supervision. But through communication, a model can align its representations for different languages *with the same shared environment*, which will thereby promote alignment between the languages themselves. A schematic overview of this paradigm can be seen in Figure 1. The code for reproducing the results reported here may be found at `https://github.com/CLMBRs/communication-translation`.

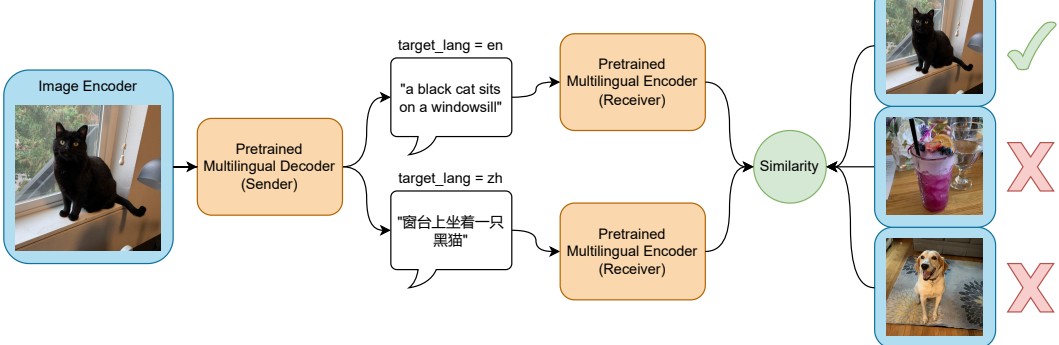

Figure 1: Learning to translate by learning to communicate: a standard image reference game, but with the sender and receiver initialized from a pretrained multilingual decoder and encoder. The communication language alternates between several of those on which the model was pre-trained.

We have operationalized some of the above choice points as follows. (Appendix A has more details.) We use mBART-25 (Liu et al., 2020), a multilingual seq2seq model that has shown strong results on unsupervised translation via the use of backtranslation (Sennrich et al., 2016; Lample et al., 2018), in a standard image reference game.[5] For the environment-to-sender adapter, we extract the final hidden layer of a pretrained ResNet, use a single linear layer to transform its dimensionality to that of mBART (the same linear transformation is used to present the images to the receiver as well) and finally use a small LSTM to generate a bag of representations for cross-attention. This adapter is trained via a small amount of (monolingual) supervised image captioning before EC. For the receiver-to-image adapter, we take the receiver's representation of a `[CLS]` token prepended to its input sequence; we also train this token via light supervision on choosing the correct image from a gold caption at the same time as the sender's adapter. Our main loss is the same as Lee et al. (2018); Li et al. (2020), where logits for image choice are inversely proportional to the MSE between image representations and the receiver's encoding of the sender's text. Pilot results exhibited

---

[5]The image reference game, as used in much of the EC literature, is very similar training an image caption module to produce discriminative captions via self-retrieval, as pursued in Liu et al. (2018). They first train the text-to-image pipeline from gold captions, and then pursue training a caption generator via image selection both with and without supervision from gold captions. We thank an anonymous reviewer for calling our attention to this work.

significant drift, with EC causing significant repetition of content words (as observed by Lazaridou et al. (2020)); adding KL regularization (Havrylov & Titov; Baziotis et al., 2020) with an mBART decoder lightly finetuned for multilingual causal language modeling mitigated the problem.

| Model | Mean (en ↔ zh) | en→zh | zh→en |
|---|---|---|---|
| mBART + BT | 10.02 | 12.45 | **7.58** |
| mBART + BT + EC | **11.01** | **14.75** | 7.27 |

Table 1: BLEU scores on WMT '19 validation data (Barrault et al., 2019).

Table 1 reports promising pilot results (see Appendix A for details): adding EC to a baseline model trained with backtranslation produces a substantial increase on average BLEU in zh↔en unsupervised translation (with largest effect on the en→zh direction). The small drop in performance with EC in the zh→en direction could be due to the small initial 'supervised' training phase which uses only English captions (see Appendix A for more details). In this phase, the decoder is being trained to produce English captions, which come from a very different distribution than the target English text for translation. This suggests experiments where the initial supervision phase comes from a different language than those used in the BT and EC phases. This could be English captions with non-English language pairs, or non-English (e.g. French) captions in our current pilot setup.[6]

These pilot results suggest that the EC-FT paradigm can produce results on a core NLP task: unsupervised neural machine translation. We are in the process of extending these proof-of-concept results along several axes—e.g. more language pairs, crucially including low-resource languages, more data / longer training, full ablations—and will write a full paper detailing all of them in the near future.

## 4    RELATED WORK

**EC and NLP**    Several other works have investigated the use of emergent communication for NLP tasks. For example, an early series of works in dialog generation use methods very similar to EC, using goal-based reinforcement learning instead of or in addition to supervised learning from existing dialogs (Li et al., 2016; Das et al., 2017; Lewis et al., 2017). Two papers have focused, in particular, on the connection between EC and machine translation. These provide much inspiration for the EC-FT approach, but do not leverage large-scale pretrained language models in the EC context.

Lee et al. (2018) use EC together with image captioning to build unsupervised translation models, showing that EC promotes better translation than the multimodal alignment technique of Nakayama & Nishida (2017) (though the unsupervised setting still lags behind supervised translation). The EC-FT approach to UNMT above differs in several important respects: we initialize our EC game with *pretrained language models*; we integrate EC with backtranslation; and we do not simultaneously train on the EC objective and image captioning objective. Moreover, because we use one multilingual model, our caption grounding only uses one language, instead of all languages. Our pilot EC-FT results suggest that the advantages of using EC for UNMT found by Lee et al. (2018) can be had even in the context of large-scale pretrained multilingual models together with backtranslation.

Li et al. (2020) use emergent communication as a pretraining step for NMT systems. That is: they have agents play an EC game, and then use those parameters to initialize an NMT system. They find that (together with adapters and weight-distance regularization) EC pretraining improves in BLEU over a standard NMT baseline, with especially large gains coming in the few-shot (500 or 1000 examples) setting. While this work shows that EC can provide a good initialization for recurrent NMT system, our pilot results suggest that EC can provide a good fine-tuning signal for a transformer-based pretrained multilingual language model in the unsupervised NMT setting.

Our approach further differs from these two in using one multilingual model, controlled by language ID codes, instead of separate models for each language.

---

[6]Thanks to an anonymous reviewer for this suggestion.

In a related vein, Lee et al. (2019) cast translation as a communication game with a third pivot language as the latent space in order to study (i) drift from a pretrained supervised MT model and (ii) using visual grounding (via gold image captions) plus language modeling to counter such drift. This approach thus does use EC with a pretrained model, but a small model trained on the target task (translation). Our approach encourages using EC in conjunction with large-scale pretrained language models which are intended to be general-purpose.

Finally, Lazaridou et al. (2020) study various ways of combining EC with a standard vision-language task, namely image captioning. As discussed in Section 2, they identify several forms of language drift and explore various ways of combining the two forms of loss. This work heavily inspired our own, since many of their settings correspond to using a pretrained image-caption system. Our focus, however, has been on using EC to fine-tune large-scale pretrained models, which introduces its own challenges and has its own benefits.

**Multimodal Pretraining** Recently, efforts in multimodal pretraining are surging, especially in vision-language (V-L) pretraining (Du et al., 2022). Most of the works create joint V-L representation through fusion encoder (Li et al., 2019b;a; Tan & Bansal, 2019), where the fused representation is the joint representation of image and text with one encoder. Other recent works also attempt using different encoders for images and texts to make the framework more efficient, such as CLIP (Radford et al., 2021) and ALIGN (Jia et al., 2021). While this previous V-L pretraining work models image and text data jointly (Du et al., 2022; Wang et al., 2021), we first use the existing pretrained language model and further pretrain it through the communication process in the referential game. Although we expect the alignment between image and text to arise through this process, we view the visual modality as an additional signal to ground the multilingual communication process.

We also note that most previous works on V-L pretraining evaluate solely on vision or V-L tasks (Li et al., 2019b; Radford et al., 2021; Jia et al., 2021), while the advantage of this joint pretraining for language-only tasks remains unclear (Yun et al., 2021; Pezzelle et al., 2021). We therefore focus on language tasks, specifically machine translation, as a way to examine the advantage of emergent communication for using the visual modality as an additional signal after pre-training.

Finally, we note that EC-FT is in a sense strictly more general than typical approaches to multimodal pre-training. While the image-based referential game used in the case study works by promoting multimodal alignment, the range of possible communication tasks that can be used in EC-FT is huge, from directing other agents in an environment (Mordatch & Abbeel, 2018) to controlling a robot (Das et al., 2019) to playing various types of games and reasoning about social dilemmas (Jaques et al., 2019). This wide range of tasks can incorporate many dimensions of communication that should be beneficial for NLP systems—e.g. other agents with their own private goals, social context, embodied control—that are not easily captured by multimodal pretraining (Bender & Koller, 2020; Bisk et al., 2020). In terms of Bisk et al. (2020)'s *world scopes* mentioned in the introduction, multimodal pretraining corresponds to world scope 3 (perception); EC-FT has the ability to move us much closer towards the final scopes 4 (embodiment and action) and 5 (the social world).

**Multimodal Fine-tuning** A related body of work focuses on adapting pretrained language-only models for use in multi-modal tasks. For example, Tsimpoukelli et al. (2021) show that using a frozen language model and adapting a visual encoder to produce embeddings aligned with the LMs' can be useful for few-shot learning in multimodal tasks like visual question answering. Liang et al. (2022) make this approach more modular by additionally freezing the visual encoder and learning separate prompt vectors. In the EC-FT context, these works suffer some of the same limitations as multimodal pretraining discussed above, but could provide very useful methods for the Environment-to-sender adapter step discussed in Section 2.

## 5 CONCLUSION

We have articulated the EC-FT approach of using emergent communication to finetune pretrained language models for use in text-only NLP tasks and illustrated this approach via a proof-of-concept on unsupervised machine translation. This general approach has the possibility of leveraging the benefits of large-scale pretraining while overcoming some of that approach's limitations by embedding pretrained models in goal-directed, interactive communicative scenarios. Much future work remains to be done: we hope that others will be encouraged pursue it.

ACKNOWLEDGMENTS

We thank Emily M Bender, Emmanuel Chemla, Chris Potts, Tania Rojas-Esponda, and the anonymous reviewers of this workshop for helpful discussion. This work was partially supported by funding from the University of Washington Royalty Research Fund, under the project "Learning to translate by learning to communicate".

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

## A  TRAINING DETAILS FOR PILOT STUDY

We here include more details about the training protocol for the pilot results reported in Section 3. As mentioned there, a more detailed and exhaustive set of experiments will appear in a future paper.

**Pretrained model**  We start from mBART 25 (Liu et al., 2020).

**BT data**  Following the mBART paper, we use the zh/en portions of the Common Crawl 25 dataset, which is the same data used for its pretraining, to generate synthetic data in backtranslation.

**Evaluation data**  We evaluate using BLEU on the WMT'19 news translation shared task data for our two language directions (Barrault et al., 2019), which is in fact the test set for the news translation task from WMT'18. We generate using beam search with a width of 5.

**mBART + BT training**  Our baseline UNMT model is trained using on-the-fly iterative backtranslation. We do 1536 steps for each direction, using a batch size of 32 and a maximum generated sequence length of 64. We use a learning rate of $1 \times 10^{-5}$ with a linear warmup of 192 steps. Inspired by Liu et al. (2020), we constrain generation: for the first 384 steps, the model can only generate tokens in the desired language among the 90% most frequent; afterwards, this is lifted to the 99% most frequent tokens in the language. (This helps prevent the model from copying the source text.) The best model is chosen by cross entropy loss on the WMT '19 validation set, evaluated every 64 steps.

**mBART + BT + EC training**  We aim to show that EC training can improve UNMT performance in the setting of using a large pretrained model together with BT. To that end, we use the same amount of BT, but additionally train on an image reference communication game in the middle. In particular, we have the following pipeline:

1. 512 steps of backtranslation: following the protocol as above, but with no linear warmup of the learning rate.

2. Supervised image captioning. In order to provide initial training signal to the various small adapters, we do a small amount of supervised training.

   We train the sender (mBART decoder + image adapters) on 10,000 English image captions from MS-COCO (Lin et al., 2014). At the same time, we train the receiver to choose the original image from $k = 15$ distractors on the basis of the gold caption, prepended with a special CLS token. The sender uses standard cross-entropy loss; the receiver uses the same loss as in (Lee et al., 2018; Li et al., 2020).

   We do both of these for 2048 steps, using the same learning rate schedule as in BT, with batch size of 16.

   Following Lee et al. (2018); Li et al. (2020), image features are extracted from ResNet 50 He et al. (2016). These 2048 feature-vectors are mapped to the 1024-dimensional mBART hidden size using a single linear layer, which is shared between sender and receiver.

   On the sender side, we take this 1024-dimensional output, and unroll it into a sequence of length 32 using a single-layer LSTM. The sequence of representations generated by this LSTM is used for the cross-attention in the sender in place of the encoder representations it saw during pretraining.

3. EC training. We do 1024 steps of EC training on the image reference task. The images also come from MS-COCO, but are ensured not to overlap with the ones used in the previous

supervision step. We used the same number of distractors and image selection loss as described above, using the straight-through Gumbel-Softmax trick to backpropagate through the sender's generation (Jang et al., 2017; Maddison et al., 2017).

In addition to the image-selection loss, we add KL regularization (Havrylov & Titov; Baziotis et al., 2020) with a language model, using $\lambda = 0.125$ as a weight for this term. In particular, we fine-tuned mBART's decoder only, with cross-attention entirely turned off, as a multilingual causal language model on 500k examples from the CC25 corpus.

4. Final BT. We finish with a remaining 1024 steps of BT. This has a constant learning rate, and the 99% mask.

Thus, steps (1) and (4) together are the same BT training protocol as our baseline; but we have added steps (2) and (3).

