# OpenReview forum: "Emergent Communication Fine-tuning (EC-FT) for Pretrained Language Models"
_ICLR.cc/2022/Workshop/EmeCom — EmeCom Workshop at ICLR 2022_

### Official Review · Reviewer_GzLG · 2022-03-21
**Beware of the literature**

**Rating:** Weak accept
**Confidence:** 5

**Review:**

It is quite dangerous to start a paper this way:
"It has recently been argued that the currently dominant paradigm in NLP of pretraining on text-only corpora will not yield robust natural language understanding systems."
The core reasons of this argument is based on the symbol grounding problem defined in 1990... https://arxiv.org/html/cs/9906002
So, there is indeed an inherent limitation of text-only system by design.
The correct framing would here: to what extend the symbol grounding problem may limit the results of language understanding. Is it a true glass ceiling, or its impact may be effectively circumvented.
Please be carful in your claims/statements... it really give a negative perspective on the first line

So... as a first comment, I would recommend the authors to mention the symbol grounding problem in the introduction, as it raises the debate about the importance of multimodal (and/or embodied) language systems. Similarly, 95% of your citations are after 2019 (i..e 3years old<) while (simulated) emergent communication is more than 30years old.  On one side, it is good to have a snapshot of the recent literature, but you. cannot discard past works. Neural networks were already tried in the 00's (https://science.umd.edu/faculty/wilkinson/Wagneretal03.pdf).

In such a positional paper, it is really crucial to be as complete as possible in the bibliography, and being extremely careful in how the field perceive the topic.

On the one hand, using language games to test large language models is appealing. On the other hand, I am not sure to understand the novelty behind EC-FN. As far as I could tell, the authors perform a task-finetuning on diverse tasks, e.g. image retrieval. Yet, multiple similar approach were done in other tasks, e.g. Referit, GuessWhat, Visual Dialog etc. While here, the task is slightly as it is image captioning with image retrieval (which has also been explored https://openaccess.thecvf.com/content_ECCV_2018/papers/Xihui_Liu_Show_Tell_and_ECCV_2018_paper.pdf).

Overall, this paper provides an interesting snapshot about the recent literature of EC and large language models. However, it lacks quite. some perspective on the original research questions, past EM works, and computer vision works. Hence, I can only encourage the authors to come to. the. workshop to have a more concrete perspective of the field.

---

### Official Review · Reviewer_6TXB · 2022-03-21
**Very interesting proposal, well written**

**Rating:** Accept
**Confidence:** 4

**Review:**

This work proposes to finetune pretrained language models using the EC paradigm. Though the idea itself is not new, this work uses a larger model than previous and focuses on the practical application of unsupervised translation with image captioning as the EC task. The literature review and introduction is quite clear (although missing some keys points) and motivation is excellently written and I appreciated the references to Bender and Koller as well as Bisk et al. The method of introducing EC into the BT pipeline is unique and though the experiments themselves are not close to SOTA, they do show promise with 1 BLEU being a significant margin for unsupervised translation. I definitely recommend this paper for the workshop and believe it will make for excellent discussion. I leave constructive criticism and helpful notes below so that the paper may be improved for possible later publication.

Experiments
- It is notables that the EC-FT method slightly decreases the accuracy of ZH-EN and that the sender is only supervised pretrained on english captions. Is it possible that the supervised english captions are the reason for this? If the authors could pretrain the model on the same MSCOCO dataset with french captions and demonstrate the same improvement it would be an even stronger result.
- Though the approach may be general, this paper seems like an interesting way to use monolingual image caption data to improve translation. If the authors wish to take a more general view, perhaps they can show that this method can work to leverage a variety of multimodal monolongual data.
- I would be interested to see experiments with iterated back-translation and iterated BT+EC-FT


Related Work
- one of the first papers on using EC-like setups for optimizing pretrained language models is missing "Deal or no Deal.." Lewis et al
- Lee et al (2018) as well as Lu et al (2020) both use pretrained language models in the form of LSTMs pretrained on IWSLT. It is true that you are using larger models and transformers which may be novel
- there is a literature on multimodal finetuning that I believe is more relevant to your work than multimodal pretraining. Specifically works like  "Multimodal Few-Shot Learning with Frozen Language Models" may be helpful in specifically defining the way that your models are finetuned

---

### Comment · Program_Chairs · 2022-04-01
**Runner-up Best Paper**

After going through all accepted papers, the program chairs have decided that this work is one of the best 3 papers at the workshop and a runner-up for the best paper award. We would like to congratulate the authors!

We found the idea of fine-tuning pretrained models using emergent communication to be an interesting and under-explored direction. Furthermore, the idea of using emergent communication as a task for unsupervised translation to be both very novel and surprisingly effective in preliminary experiments. This work opens up a "new frontier" of grounding large pretrained models in tasks as well as finding out which emergent communication tasks would be the most beneficial for unsupervised/self-supervised training. We look forward to seeing this work extended and built upon!

---

### Decision · Program_Chairs · 2022-03-25

**Decision:**

Accept

**Comment:**

Both reviewers found this work worthy of being accepted and despite some possible issues with literature they agree that it will spark interesting discussions at the workshop